# Perylene Imide-Based Optical Chemosensors for Vapor Detection

**Miao Zhang [1], Jiangfan Shi [2], Chenglong Liao [1], Qingyun Tian [1], Chuanyi Wang [1], Shuai Chen [2,3,*] and Ling Zang [2,*]**

[1] School of Environmental Science and Engineering, Shaanxi University of Science and Technology, Xi'an 710021, China; miaozhang1209@163.com (M.Z.); lcl6862@163.com (C.L.); tianqingyun2017@163.com (Q.T.); wangchuanyi@sust.edu.cn (C.W.)

[2] Nano Institute of Utah and Department of Materials Science and Engineering, University of Utah, Salt Lake City, UT 84112, USA; jfan_shi@163.com

[3] Flexible Electronics Innovation Institute, Jiangxi Science & Technology Normal University, Nanchang 330013, China

* Correspondence: shuai.chen@utah.edu (S.C.); lzang@eng.utah.edu (L.Z.); Tel.: +1-801-587-1551 (L.Z.); Fax: +1-801-581-4816 (L.Z.)

**Abstract:** Perylene imide (PI) molecules and materials have been extensively studied for optical chemical sensors, particularly those based on fluorescence and colorimetric mode, taking advantage of the unique features of PIs such as structure tunability, good thermal, optical and chemical stability, strong electron affinity, strong visible light absorption and high fluorescence quantum yield. PI-based optical chemosensors have now found broad applications in gas phase detection of chemicals, including explosives, biomarkers of some food and diseases (such as organic amines (alkylamines and aromatic amines)), benzene homologs, organic peroxides, phenols and nitroaromatics, etc. In this review, the recent research on PI-based fluorometric and colorimetric sensors, as well as array technology incorporating multiple sensors, is reviewed along with the discussion of potential applications in environment, health and public safety areas. Specifically, we discuss the molecular design and aggregate architecture of PIs in correlation with the corresponding sensor performances (including sensitivity, selectivity, response time, recovery time, reversibility, etc.). We also provide a perspective summary highlighting the great potential for future development of PIs optical chemosensors, especially in the sensor array format that will largely enhance the detection specificity in complexed environments.

**Keywords:** chemosensor; vapor detection; perylene imide; fluorescence; colorimetric

## 1. Introduction

Developing chemosensor techniques for trace-level detection of vapor analytes, especially volatile organic compounds (VOCs), and nitro explosives, which are of various health and security concern, has drawn increasing research interest and efforts in the past decades [1–3]. As we know, conventional analytical instrumentations such as gas chromatography (GC), mass spectrometry (MS), infrared spectroscopy (IR), high-performance liquid chromatography (HPLC) [4–8], as well as some sensor methods such as surface acoustic wave (SAW) sensors [9], quartz crystal microbalances [10] and electrochemical resistors [11] are normally managed in lab detection by professional and technical staff due to their expensive cost, bulk size or complicated and time-consuming operation. In contrast, chemosensors such as electrochemical sensors, field-effect transistor sensors, chemiresistive sensors, fluorometric and colorimetric sensors exhibit superiority regarding system simplicity for facile operation, and non-destructive detection with high sensitivity and selectivity [12–14]. Among these chemosensing methods, optical sensors (including fluorometric and colorimetric types) are highly attractive for chemical vapor detection not

only for real-time operation, compact size, immunity to electromagnetic interferences and remote sensing capabilities, but also for simple readout of output signals (sometimes just by naked eyes). These vapor chemosensors are usually fabricated from fluorophores or chromophores as active sensing materials (also known as probes) in the form of solid films, relying on the rational design of the molecular structures. Compared to those molecule or particle-based sensors in solution-phase detection, these sensing films are usually prepared by immobilization of fluorescent or colorimetric sensors (in molecule, particle or fibril form) on suitable solid substrates, providing advantages in terms of their reusability and reproducibility, which are two crucial factors for practical applications and actual device development of chemosensors [14–16].

Among the widely diverse optical sensor materials studied thus far, perylene imides (PIs) have attained significant attention in the past decades, because PIs are exceptional n-type organic materials with excellent electron affinity, strong fluorescence (high quantum yield), strong visible light absorption, high thermal stability and photostability under ambient conditions. Indeed, the combination of these unique features makes PIs ideal materials for development in various electronic and optoelectronic devices including solar cells, light-emitting diodes, photoelectric sensors and chemosensors [17,18]. For example, PIs-based nanofiber films have been extensively studied by our and others' groups for sensing a wide range of reducing gaseous species (i.e., electron-donating), such as ammonia ($NH_3$) and volatile organic amines, by measuring the change of fluorescence intensity upon exposure to the analyte vapor [17,19]. The key of PIs-based chemosensors for successful vapor detection is introducing suitable functional groups on the imide or bay position to enable strong specific binding towards target analytes, while still maintaining the active fluorescence response and certain solubility for solution processing [17,20]. The two kinds of PIs widely used in the field of chemosensors are perylene diimides (PDIs) and perylene monoimides (PMIs), with most research focused on PDIs mainly due to their facile synthesis and good solution processability. However, PMIs usually retain the strong fluorescence upon assembly (aggregation) into solid state, while the solid state of PDIs is normally very weak in fluorescence mainly due to the p-p stacking quenching. In view of the primary requirement of sufficient fluorescence for sensor application (particularly those based on fluorescence quenching mode), PMIs provide some unique opportunities for sensor development, complementary to PDIs. Specifically, the anhydride group at one side of PMIs is intrinsically a strong and specific binder to amines through hydrogen bonding in concert with donor-acceptor interaction [17]. In 2008, our group reported on a porous fluorescent film chemosensor for the vapor detection of aniline; the film was composed of intertwined nanofibers assembled from a PMI molecule (PMI-1 shown in Figure 1) [21]. The successful fabrication of shape-defined, highly crystalline nanofibers was mainly driven by the one-dimensional (1D) intermolecular arrangement, while the tilted p-p stacking still allows for sufficient fluorescence in the solid state. Upon surface adsorption of amines (electron donor) like aniline, the fluorescence of PMIs gets efferently quenched through photoinduced electron transfer (PET). Such fluorescence quenching based sensors have proven highly sensitive for vapor detection of amines with limit of detection down to ppb level for aniline, taking advantage mostly of the unique nano-porosity of the nanofiber film, wherein the intertwined nanofibers form a porous "super net" structure with a large, open surface area that is conducive to both molecular diffusion and adsorption. The innovative materials design, along with specific nanoscale engineering, opened a new, promising direction for the development of solid-state fluorescent chemosensors [21], as evidenced from many follow-up studies in developing various PIs-based fluorescence sensors for chemical vapor detection [22–25]. Table 2 lists some representative PDIs and PMIs that have been developed and proven effective chemosensors for vapor detection in recent years.

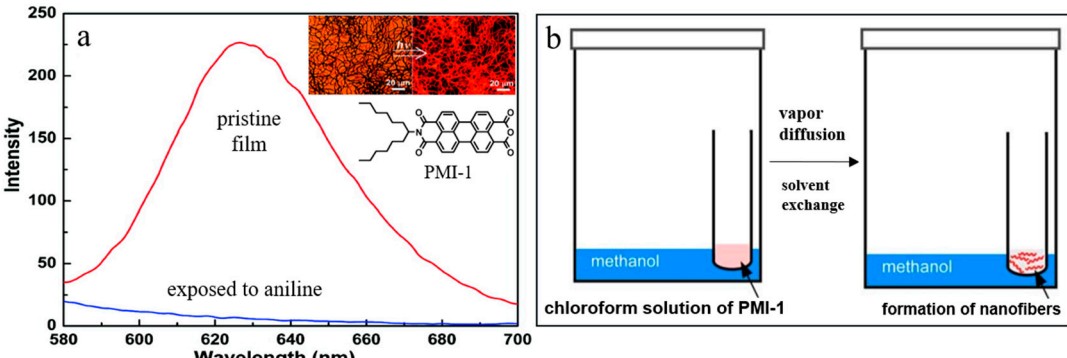

**Figure 1.** (**a**) Fluorescence spectra of PMI-1 nanofibril film before (red) and after (blue) exposure to the saturated vapor of aniline (880 ppm) for 10 s. Inset shows bright field and fluorescence optical microscopy image of the film; (**b**) A diagram showing the procedure of fabrication of the nanofibers via slow solvent vapor transfer [21].

So far, some reviews have been published by our group and others on the solid phase PDI-based chemosensors, which are capable of both gas and solution phase detection. For example, Fu, Y [22] reported on a review mainly focused on the detection of organic amines, and the associated sensing mechanism and the design principles of probes. Fang's group [26] summarized the sensory materials and device fabrication. Our group [19] recently published a review on the recent advance in the development of PDI-based optical sensors for environmental detection of heavy metal ions, inorganic anions, amines and other organic pollutants in aqueous and other liquid phase solutions. The PDI sensors covered in the previous review are mostly based on molecular state and the sensing response is caused by molecular binding or reactions in liquid solutions, such as ion coordination or complexation, protonation/deprotonation or other chemical reactions. In sharp contrast, the sensors covered in this review are based on solid aggregate of PDIs and work for gas phase detection of chemicals. We also published another review specifically focused on the nanoscale structural control of PDIs nanofibril interfacial heterojunctions via molecular self-assembly, with the aim to optimize the chemiresistive sensing performance so as to achieve trace level chemical vapor detection [27]. Meanwhile, Kumar, K. and co-workers [28] published a specific review on the aesthetic design strategies and sensing mechanism of PDI-based sensors. However, the field still lacks a comprehensive review on optical chemosensors covering both PDIs and PMIs-based sensor materials, which combined would enable much broader detection of chemical vapors beyond the common species like amines. Moreover, multimodal sensors and array systems involving multiple sensor components have not been adequately reviewed for PDIs or PMIs sensors, though the multiplex sensor systems have recently drawn increasing attention in chemosensors, which help enable or enhance the detection capability for complex or mixture samples with the assistance of algorithm data analytics like machine learning.

In this review, we provide a comprehensive overview of the recent progress of PIs-based fluorometric and colorimetric sensors, covering both the sensing system and device design. Both PDIs and PMIs are discussed and compared for the sensor performance regarding sensitivity, selectivity, response time, reversibility, etc. Many of these sensors show successful detection of a wide range of chemical vapors including organic amines (both alkyl amines and aromatic amines), benzene homologs, organic peroxides, as well as phenols and nitroaromatics. Specific efforts of this review are put on analysis of the molecular design and structural engineering, and the correlation with the sensing mechanisms and performances, with a hope to provide further guidance for the design and optimization of new sensor materials based on PIs or other related molecules or composites. To this end, a perspective summary is given at the end of the review discussing both the key challenges and potential solutions for the future research and application development in optical chemosensors.

## 2. Physicochemical and Optical Properties of PIs

Bulk-phase PIs have been widely used in various optoelectronics devices [29–31], including thin-film transistors [32,33], photovoltaics [34,35], liquid crystals [36,37] and chemiresistive sensors [38], etc. Performance of optical chemosensors (as well the devices fabricated therefrom) depends primarily on the physicochemical and optical properties of the sensing materials, here specifically the fluorophores or chromophores, which in turn are tightly correlated with the molecular design and structure engineering [28,29,39]. For PIs-based materials, there are three unique features that render their excellent applications in optical chemosensors (especially for vapor detection) [7]. Firstly, the PI molecule possesses a large, planar and rigid π-conjugation system based on the central perylene skeleton, which is of π-electron deficient aromatic nature, making it a strong electron acceptor (n-type) with good environmental chemical stability. Due to the highly efficient p-p transition, PIs exhibit strong visible light absorption and fluorescence (close to unit quantum yield in good organic solvents), well suited for application in optical sensors based on fluorescence or colorimetric modulation [19]. Secondly, molecules of PIs are flexible for structural modification through facile organic synthesis by changing the substitution at the imide position or bay area, which in turn can tune the solubility, spectroscopic and redox properties, as well as the self-assembly of PIs. Moreover, the imide (nitrogen) position on PI is a node in the π-orbital molecular wavefunction, meaning that modification with different substitutions at the position will not change the electronic (light absorption and fluorescence) properties. This provides a number of viable pathways for controlling the molecular structure of PIs so as to achieve the best sensor performance. Nonetheless, the bay-area substitution usually brings significant changes of the optical and electronic properties of PIs due to the direct interruption to the π-conjugation [40]. Lastly, PIs are one of the most studied classes of n-type semiconductor molecules for self-assembly that form structure and morphology defined nanomaterials [17]. Particularly, owing to the planar rigid π-conjugation geometry, PIs favor one-dimensional assembly, forming shape-defined nanofibers driven by the strong π-π stacking between perylene planes. There are enormous options for the one-dimensional molecular assembly to be optimized for both structure and morphology through changing the side substitutions at the imide or bay positions. The co-facial π-π stacking (in the format of *H*-aggregate) results in fluorescence quenching and a hypsochromic shift of the absorption [21,41]. While the nanofibers of PIs have been extensively studied as optical and chemiresistive sensors [17,23,27], the aggregation-disaggregation induced fluorescence change of PIs have also been developed into fluorescence sensors for detecting inorganic or organic ions in liquid-phase environment [19]. However, for vapor analytes sensing, PIs have to be fabricated in the form of films or other formats of solid substrate, for which it is imperative to retain sufficient fluorescence of PIs in solid state (i.e., minimizing the fluorescence quenching caused by π-π stacking). In most cases, significantly increased fluorescence can be achieved by introducing large bulky moieties at either the imide position or the bay area; bulky moieties provide steric hindrance, thus tilting the co-facial π-π stacking, which in turn weakens the fluorescence quenching effect [20,42]. In addition, it has been demonstrated by our group and others that the construction and performances of PIs-based optical chemosensors are closely related to the different dimensions and morphologies of PIs assemblies [21,43–46]. For example, porous films with 1D PIs nanofibers showed significantly enhanced sensing performances in comparison to those with irregular morphologies. As mentioned above, relying on the good tradeoff between ordered molecular assembly and retaining strong fluorescence in solid state, our group initially explored the PMI-based fluorescent nanofiber sensor for trace vapor detection of organic amines [21]. Following the initial work, Che and co-workers [47] have successfully fabricated thermally stable bilayer nanocoils based on an asymmetric PDI molecule, which demonstrate great sensor performance in detecting amines like aniline and phenethylamine.

Combination of the excellent physicochemical and optical properties mentioned above makes PIs an ideal fluorophore and chromophore materials for being developed as efficient chemosensors that are capable of vapor defection, targeting a wide range of environmental

pollutants, explosives and biomarkers of some food and diseases, including organic amines, BTEX (benzene, toluene, ethylbenzene, *o*-xylene, *m*-xylene and *p*-xylene), organic peroxides, phenols, nitroaromatics, etc. Beside molecule design and synthesis, many efforts have been laid on the fabrication (and nanoscale engineering) of optically active thin (often porous) films from PIs. The sensor performance against environmental factors such as temperature, humidity and concentration, and in correlation with the thermodynamic parameters of analytes like boiling point, has also been studied in detail, aiming to further improve the robustness and sustainability of sensor system as desired in practical use [23]. The environmental factors may also affect the intermolecular association of PIs aggregates, thus causing change in sensing performance [48,49].

### 3. Fabrication of PIs-Based Optical Chemosensors

Optical sensors fabricated from PIs are mainly developed in the aspects of fluorescent film construction, array technology and sensing platform. There are many methods for constructing thin-film based fluorescent sensors as described in this review, such as physical coating technology [42], Langmuir-Blodgett (LB) technology [50] and molecular gel technology [51]. Among them, physical coating technology, including spin coating, dip coating and spraying, is the most commonly used method for constructing fluorescent thin film due to its easy operation [41]. However, this method requires strict instrumentation control in order to obtain high-quality films. Conversely, LB technology is relatively easy to operate and control, and the films obtained usually have highly organized structure, and to some extent, such structure can also be controlled or modified upon various needs. Molecular gels, on the other hand, mainly form three-dimensional (3D) network structure through interactions between molecules such as Van der Waals force, hydrogen bonding, π-π stacking, solvophobic vs. solvophilic interactions and so on [52]. Formation of molecular gels normally exhibits stimulus phase change (shear stimulation and temperature change stimulation); therefore, the gels can be considered as physical gels, involving small molecules as gelling agents.

As for a complete sensor system or testing platform, it is typically composed of a sensor unit (material), a sampling part including gas (vapor) supply under precise control of flow rate, a signal readout device like photodiode for measuring the light intensity and an environmental chamber holding all the parts in a closed space where temperature, humidity and other experimental conditions can be controlled (Figure 2a,b) [24]. The gas supply system is designed to work in a stationary manner to adjust sample volume or flow rate so as to control the vapor concentration mixed in ambient or zero air. The environmental chamber is usually required and critical for assessing the sensor performance against real world ambient conditions, with varying temperature and humidity, as well as presence of other interference chemicals or gases. While the sensor unit is often based on a single sensor material or sensing mode, nowadays the array-based sensing approach has become more popular and attracted increasing interest, particularly because the recent advancement in electronic circuit and on-chip data processing makes the incorporation of multiple sensors on a chip to be more feasible [53]. An array with multiple sensors or sensing modes integrated can not only selectively identify a single analyte, but also enable or enhance the capability of differentiating multiple analytes in a mixed sample, which are actually more common in real world detections. For example, a colorimetric sensor array has recently been reported that effectively discriminates ten explosives [54]. Particularly for colorimetric or fluorescence sensing mode, varying materials and molecules (e.g., dyes, nanomaterials and polymers) have been developed and incorporated into an array to target a broad range of chemical vapor analytes [55,56]. Taking advantage of the specific, strong chemical reactions (not just physical interaction) between the sensor and analytes, these sensors in an array can provide unprecedented sensitivity and selectivity in detection of different analytes within complexed or mixed samples.

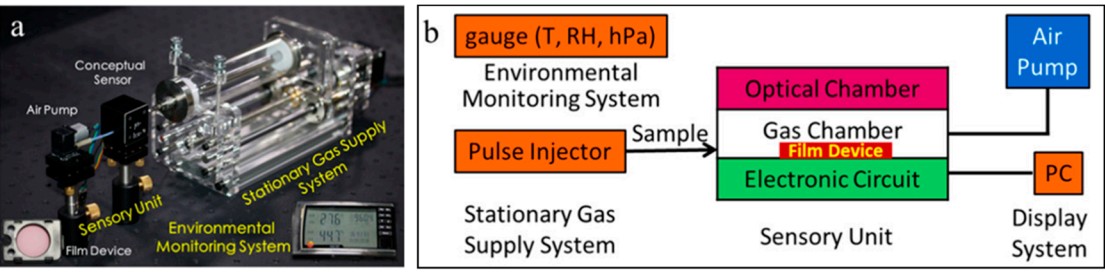

**Figure 2.** (**a**) The picture of the gaseous analyte detection platform based on the film device; (**b**) Schematic description of the home-made sensing platform [24].

## 4. PIs-Based Optical Chemosensors for Different Types of Analytes

### 4.1. Organic Amines

PIs-based optical chemosensors can respond to various organic amines in the environment, which are widely used in fertilizers, pesticides, dyes, pharmaceuticals, surfactants, food manufacturing industries and so on [57,58]. For instance, the volatile alkylamines like ethylenediamine which are commonly used in printed circuit boards and metal finishing industries are typical air pollutants [25], while trimethylamine, 1,4-diaminobutane (putrescine) and 1,5-pentanediamine (cadaverine) are critical biological amines released during the spoilage or decomposition of meat protein [59]. Importantly, aromatic amines such as aniline and toluidine are usually considered the biomarkers of diseases, particularly cancers; detection of the amines could enable diagnoses or even early screening of diseases, thus providing more options for medical treatment [60]. Inhalation of or exposure to volatile amines can cause health risks including methemoglobin disease, hemolytic anemia, and liver and kidney damage [57]. Therefore it remains imperative to detect and monitor the amines in environments, for which fluorescence and colorimetric optical sensors (based on fluorescence intensity and color change, respectively) have attracted enormous attention and research efforts due to their high sensitivity and selectivity, low cost and ease of use, as well optional design for performance improvement. Especially for PIs-based optical sensors, the intrinsic strong PET modulation makes them one of the most studied sensor materials in optical chemosensors for detecting organic amines (which act as strong electron donors) [17]. Meanwhile, the chemical interactions between the analyte and PIs often causes dramatic color change, providing an alternative way of sensing through colorimetric modulation [12]. To this case, the diverse chemical interactions (e.g., donor-acceptor interaction, hydrogen bonding, and many other non-covalent interaction) involved in colorimetric sensors make it uniquely powerful for differential sensing (especially via sensors array) that enables discrimination of wide range of analytes, including those with very close structure and chemical properties (e.g., alkyl amines). The differential sensing is also powerful in selective detection of analytes from the complex environmental background, with many interreferences.

### 4.1.1. Alkylamines

Among all the alkylamine analytes targeted by PIs-based optical chemosensors, diamines represent the most popular one for the sensor study mainly due to their good volatility that helps generate and control the vapor concentration. By modifying the bay area of PDI with two highly water soluble alkoxy oligomers, Fang et al. [25] designed and synthesized an amphiphilic PDI derivative (PDI-1, Figure 3) which consists of a PDI unit as the hydrophobic part and two alkoxy groups as the hydrophilic part attached at the bay position via 4-amino-phenyl linker, and developed it onto an amazing thin film fluorescence sensor for amine detection. The thin film (membrane) of PDI-1 was constructed by Langmuir-Blodgett (LB) technology, and the film as fabricated showed efficient fluorescence modulation upon exposure to ethylenediamine. The emission of the PDI-1 aggregate implies that the conjugation plane of PDI-1 stacks in a state far from the nonfluo-

rescent H-aggregate. Interestingly, the sensor exhibited high selectivity to ethylenediamine among diverse background of diamine vapors including cadaverine, putrescine and so on. This LB film technology provided a nice approach to precisely controlling the thickness and molecular orientation within the film. Control of intermolecular arrangement is crucial for minimizing the H-packing that normally weakens the fluorescence of PDIs. One of the remarkable unique features regarding molecular design of PDI-1 is that the two side substitutions contains large number of alkoxy (ether) moieties that enable strong binding towards amines through hydrogen bonding. Nonetheless, such strong intermolecular interactions may also make the PDIs responsive to other amines, thus causing a challenge in minimizing or avoiding the false positives of detection. Therefore, this demands further improvement of detection selectivity through materials design (with increasing binding specificity) and/or sensor array based differential sensing.

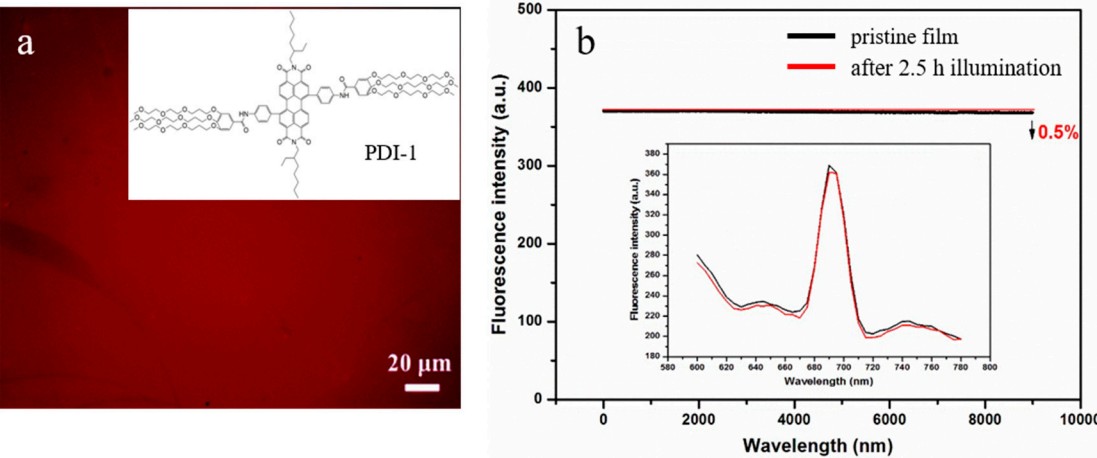

**Figure 3.** (**a**) Fluorescence microscopy image of the as-prepared PDI-1 based LB film; (**b**) Fluorescence intensity of PDI-1 based LB film (before and after 2.5 h illumination) recorded at the emission maximal wavelength 690 nm ($\lambda_{ex}$ = 460 nm, 150 W. Xe Lamp) as a function of time; the almost no decrease in fluorescence intensity (usually caused by photobleaching) indicates the extremely strong robustness of the PDI film fabricated [25]. The inset shows the full fluorescence spectra of the same film before and after 2.5 h illumination.

Additionally, as shown in Figure 3, the film of PDI-1 demonstrated superior photostability, i.e., the 2.5 h of continuous illumination resulted in only less than 0.5% reduction of the initial fluorescence intensity, which proves, on the other hand, the observed fluorescence quenching is caused by the adsorption of diamine rather than by photo-bleaching. Moreover, the fluorescence intensity of the film was measured under varying concentrations of ethylenediamine vapor, and it was found that the fluorescence quenching (sensor response) shows linear dependence on the vapor concentration within a wide dynamic range of ca. 0–8 g/m$^3$. This concentration dependent response also further proves that the fluorescence quenching was truly due to the interaction with amines. Interestingly for future practical use (especially for onsite real-time detection), the fluorescence of the film after exposure to amines can be quickly recovered simply by air blowing, indicating the reversibility of the sensor material. Notably, the fluorescence intensity of the film increases linearly with the thickness (number of layers), showing great promise for further optimization of the sensor performance regarding the tradeoff between the specific surface area and total fluorescence intensity, with the former directly determining the surface adsorption of analytes and thus the sensing sensitivity, and the latter related to the signal-to-noise ratio of measurement (in general the higher the intensity measured, the more reliable the measurement would be).

Sensitive detection of biogenic amines is crucial to the assessment of the safety and quality of meat products during storage and transportation [59]. Che and co-workers [61] developed a PI-based fluorescent chemosensor for real-time monitoring of meat spoilage.

The sensor (in a format of film) employed fluorescent nanotubes assembled from a chiral asymmetric PDI-2 molecule (Figure 4a), which demonstrated efficient fluorescence quenching response upon exposure to amines like putrescine and cadaverine (Figure 4b), two common biomarkers emitted from raw pork, chicken, fish or shrimp meat. As an example shown in Figure 4a, when exposed to the emitted amine vapor from 1 g of fresh shrimp that was placed approximately 1.0 cm from the film sensor, the fluorescence intensity decreased linearly with exposure time, with ca. 6% quenching observed after 1 h of exposure at room temperature. In contrast, the same experiment under clean air resulted in no fluorescence quenching, indicating clearly that the observed quenching with the shrimp sample present was totally due to the interaction with amines released from the degraded shrimp, and moreover the surface adsorption of amines can be accumulated with time, which is consistent with the porous morphology (and thus large open surface area) intrinsic to the nanotubes. By virtue of the inherent internal hollow structures of PDI-2 nanotubes which are conducive to the diffusion and adsorption of vapor analytes, the detection limits of putrescine and cadaverine can reach as low as 2.6 and 1.2 ppb, respectively (Figure 4c), lower than that of other volatile alkyl amines (Figure 4d). More importantly, due to the chirality induced tilting of intermolecular arrangement (with minimizing *H*-aggregate type quenching), the fluorescence quantum yield of the nanotubes remains as strong as 46%, which is much higher than that reported for other PIs aggregates (typically around or below 25%). The strong fluorescence intensity is especially favorable for development of fluorescence quenching based sensors, at least ensuring high signal-to-noise ratio that directly determines the lower detection limit.

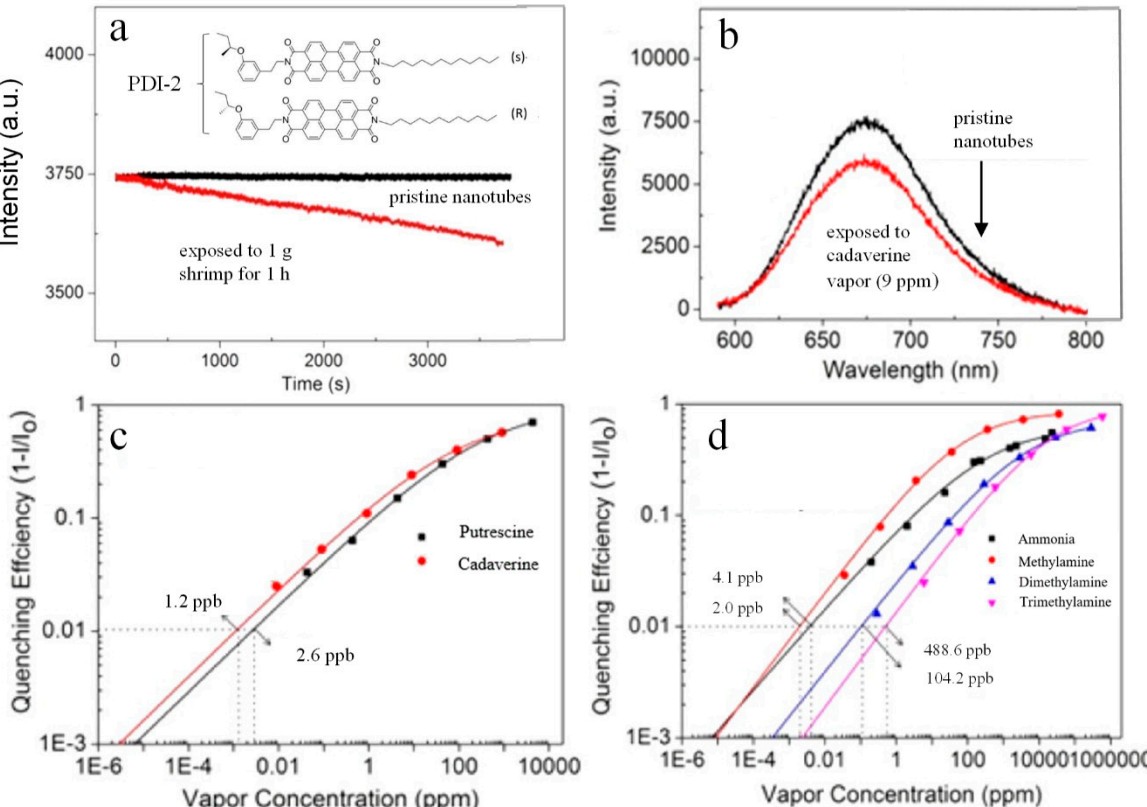

**Figure 4.** (**a**) Fluorescence intensity recorded for the nanotubes assembled from PDI-2 as a function of time at room temperature (25 °C) for 1 h, for both the pristine sample (black) and the one exposed to 1 g shrimp placed in a 1 cm proximity (red). Inset shows molecular structures of chiral molecule PDI-2; (**b**) Typical fluorescence spectra recorded for the PDI-2 nanotubes before and after exposure to 9 ppm cadaverine; (**c**) Fluorescence quenching efficiency ($1-I/I_O$) of the nanotubes as a function of the vapor concentration of cadaverine and putrescine, as well as (**d**) other amines [61].

### 4.1.2. Aromatic Amines

Unlike alkylamines, aromatic amines are not only widely used as intermediates in the chemical industry, but also play an important role in many physiological processes of organisms. Many aromatic amines are considered to be potential carcinogens by the International Agency for the Research on Cancer [60], posing a great threat to human health. For example, daily exposure to aromatic amines such as aniline and toluidine increases the risk of cancer uptake. Moreover, many illicit drugs (e.g., LSD, fentanyl) contain aromatic amines, and synthesis of the drugs often involves aromatic amines as precursors as well. Particularly, fentanyl is 50–100 times more potent than morphine, and 30–50 times more potent than heroin, and is now being used as an additive to heroin, counterfeit pills and cocaine, for illicit use. Fentanyl is extremely dangerous, and doses as small as 2–3 mg prove to be fatal. Overdoses are now the leading cause of death for Americans under 50, surpassing car wrecks, HIV/AIDS and gun deaths at their peaks. The problem of fentanyl is accelerating, becoming a national challenge for the US, with the economic cost of the fentanyl crisis in 2015 at approximately $504 billion, or 2.8% of GDP. These health and security concerns make it more critical than ever to develop efficient sensors that can detect not only the drugs themselves, but more importantly the aromatic amines as synthesis precursors, which in turn may help find and identify the clandestine processing sites. In recent years, many PIs-based optical chemosensors have been developed for quick, high sensitive detection of aromatic amines, especially aniline (the most commonly used synthesis precursor), and its volatile derivatives or analogues such as *o*-toluidine, *p*-toluidine and *o*-phenylenediamine. The research of PIs sensors has currently been drawing increasing effort from chemistry, materials science, as well as the electrical engineering related to circuit fabrication and signal processing. The increasing interest in PIs is partially driven by the unique structural feature of PIs as mentioned above, which allows for enormous options to modify the side group modification so as to tune the sensor performance through fluorescence or colorimetric signal modulation.

Sensor films of PIs have been fabricated via supramolecular self-assembly that can control both the intermolecular arrangement of PIs and the global morphology of aggregates, which combined would enable efficient sensor response towards amines vapor [41]. However, very often the fluorescence of PIs decreases significantly, becoming hard for use as sensor, upon assembly into solid state. To solve this problem, substitution at the imide position with steric, branched groups in general can help enhance the fluorescence of PIs assembly as evidenced from our previous work [21], wherein a PMI molecule (PMI-1 in Figure 1) modified with branched hexylheptyl group at the imide was designed and synthesized. Self-assembly of PMI-1 into nanofibers was performed through a slow solvent-exchange process, which was realized via vapor diffusion within a closed chamber. The branched substitutions and the asymmetric structure of PMI-1 are conducive to the formation of assembly that still maintains strong fluorescence (with quantum yield of ca. 15%). The anhydride group of PMI-1 allows for strong interaction with amines through hydrogen bonding in conjunction with donor-acceptor interaction, thus providing great promise for development as sensors for vapor detection of amines. As shown in Figure 1, a fluorescent film composed of well-defined PMI-1 nanofibers was fabricated, and demonstrated efficient fluorescence quenching upon exposure to aniline vapor. Upon deposition onto a glass substrate, the entangled nanofibers of PMI-1 (with diameter around 350 nm) formed a mesh-like film, which enabled expedient diffusion of gaseous analyte molecules within the film matrix, leading to milliseconds sensing response towards aniline vapor. Upon exposure to the saturated aniline vapor of 880 ppm, the film fluorescence was quickly quenched by almost 100% in 0.32 s. In addition, none of the other common organic liquids and solids tested (with phenol as an exception) exhibited the similar fluorescence quenching response upon exposing the PMI-1 film to their vapors. Exposure to the saturated vapor of phenol resulted in 54% fluorescence quenching, though the quenched fluorescence could be recovered almost 100% by simply re-exposing the film to the clean air for 60 min. The quick recovery of fluorescence is distinct from the case of amines, for which the quenched fluo-

rescence can hardly be recovered due to the strong anhydride-amine binding. Such distinct sensing behavior can be used for distinguishing amines from phenols and other electron donor types of analytes. In general, nanofibers assembled from building blocks like PI molecules provide a unique format of sensor materials that possess 1D enhanced optical and electronic properties along the long-axis, such as the long-range exciton diffusion, mainly due to the extensive *p-p* interaction between the stacked molecules. Long-range exciton diffusion enables amplified fluorescence quenching and thus enhanced sensing for vapor detection [43]. Moreover, the large open surface area intrinsic to nanofibers is conducive to increasing surface adsorption.

The surface area of nanofibers can simply be increased by shrinking their size (diameter), which in turn can potentially help enhance the sensing efficiency as evidenced from our follow-up work on PMI-1 [62]. Much thinner nanofibers (with diameter of only 30–50 nm) were fabricated from PMI-1 using a modified self-assembly method (Figure 5a). Under the same testing condition, the new nanofibers film demonstrated much enhanced fluorescence quenching upon exposure to the vapor of aniline in comparison to the previous PMI-1 nanofibers that were of much larger size (~ 350 nm). As shown in Figure 5c, under the same vapor pressure of 35 ppb, the fluorescence quenching of film deposited from 0.1 mg nanofibers can reach 39%, larger than 31% for using 0.15 mg or 13% for using 0.35 mg, but all better than that using 0.35 mg larger nanofibers. An increase in quenching efficiency implies direct improvement of the detection limit. By fitting the data with the Langmuir adsorption model, the detection limit level can be estimated to be 0.1 ppb (for the film of 0.15 mg nanofibers), which is about a 50 fold improvement from the detection limit of 5 ppb determined for the larger fibers of 350 nm. Considering the same driving force for the PET process (Figure 5b), the observed improvement of fluorescence quenching is mostly owing to the decrease in nanofiber size (or increase in surface area).

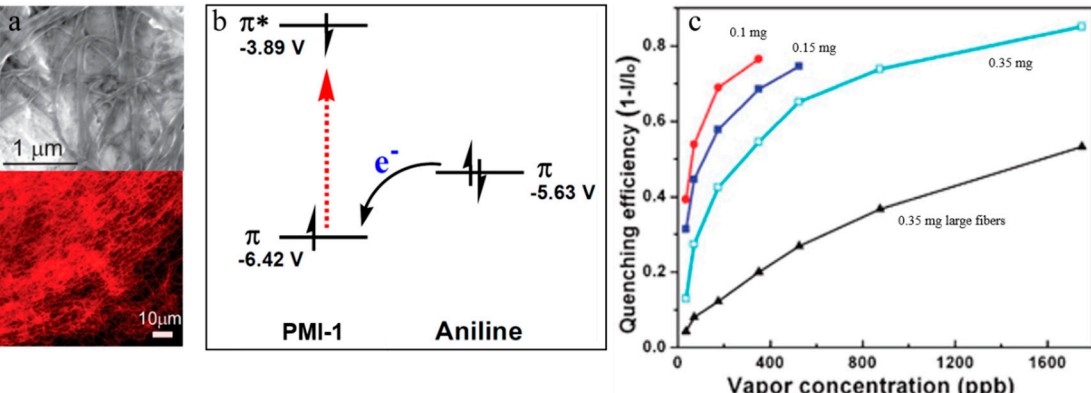

**Figure 5.** (**a**) SEM image and fluorescence microscopy image of the PMI-1 nanofibers deposited on a glass slide; (**b**) Energy levels of HOMO ($\pi$) and LUMO ($\pi^*$) orbitals of PMI-1 and aniline showing the thermodynamically favorable PET process for fluorescence quenching; (**c**) Fluorescence quenching efficiency ($1-I/I_o$) measured for the PMI-1 film (deposited from different amounts of nanofibers) as a function of the vapor concentration of aniline [62]; also included for comparison are the corresponding sensor data measured for the larger PMI-1 fibers as reported in another paper [21].

Taking the advantage of synthetic flexibility of changing the substitution at imide positions, the binding strength and specificity of PIs can be further improved by attaching special functional groups to the imide position of PIs. Particularly for the vapor detection of aniline, Liu and co-workers reported on two PDI-based molecules, PDI-3 [63] and PDI-4 [64] (Figure 6a), based on the same terminal permethyl-β-cyclodextrin group as well-defined receptor site for selective binding of aniline molecules. Different linkage chains between cyclodextrin group and PDI core were used for the two PDIs; that is, the linkage chains between cyclodextrin group and PDI-4 core were longer than PDI-3, in order to modulate the intermolecular stacking and thus the morphology of nanomaterials assembled therefrom. In view of the presence of a longer junction spacer between cyclodextrin and PDI

in PDI-4 molecules, they demonstrate one to two orders of magnitude stronger aggregation capability due to the weakened steric hindrance of cyclodextrin to the PBI π-stacking. Benefiting from the side-group modification of cyclodextrin, PDI-3 and PDI-4 exhibit good water solubility, which facilitates the self-assembly process in water solution under varying concentrations. Both PDI-3 and PDI-4 aggregates show nanorod morphology and solid-state red fluorescence emission. Figure 6b,c show the well-ordered nanorod structure of PDI-3 as imaged by scanning electron microscopy (SEM) image and the strong red emission imaged by fluorescence microscopy.

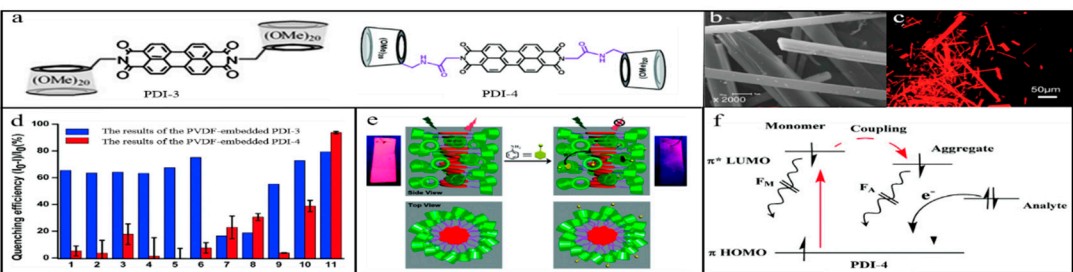

**Figure 6.** (**a**) Molecular structures of PDI-3 and PDI-4; (**b**,**c**) SEM and fluorescence optical image of PDI-3 nanorods; (**d**) Fluorescence quenching efficiency measured over the PVDF-embedded films of PDI-3 and PDI-4 upon 10 s of exposure to the saturated vapors of various amines, nitro-based compounds, and general organic solvents: 1, toluene; 2, methanol; 3, chloroform; 4, nitromethane; 5, nitroethane; 6, acetonitrile; 7, hydrazine hydrate; 8, benzylamine; 9, triethylamine; 10, butylamine; 11, aniline. Error bar: standard deviation of measurement; (**e**) Schematic illustration of the aggregation morphology of PDI-4 on PVDF and its sensing process for aniline; (**f**) Energy level diagram showing the direction and driving force for PET between analyte and PDI-4 in both molecule and aggregate state [63,64].

To display the potential application of PDI-3 and PDI-4 nanorods in chemical vapor detection, the self-assembled materials were deposited on a film of polyvinylidene fluoride (PVDF) via a solution process. Briefly, a PVDF film was immersed in the aqueous solution containing PDI-3 or PDI-4, followed by air drying, leading to formation of aggregates (assembled nanorods) directly embedded on the surface. The film-supported nanorods thus prepared are suited for exposure to chemical vapor and testing the sensor response simply by measuring the change in fluorescence emission. As shown in Figure 6d, film of PDI-4 demonstrated significant stronger fluorescence quenching response towards the vapor of aniline, in comparison to other chemical vapors tested under the same conditions. This is dramatically different from the responses observed for PDI-3 also shown in Figure 6d, implying that a combination of sensors PDI-3 and PDI-4 would enable selective detection of aniline. The observed strong fluorescence sensing response is likely due to the enhanced vapor capture by the cyclodextrin cavity (Figure 6e). The captured aniline acts as a strong electron donor that can quench the fluorescence of PDI via a PET process. The well-ordered columnar stacking of PDIs allows for efficient migration of the photo-excited (fluorescent) state along the long axis of nanorods, thus enabling amplification of the fluorescence quenching [43], i.e., a captured aniline may quench the fluorescence of any PDI within the stacks in proximity (Figure 6e,f). Moreover, the encapsulation of aniline into the cyclodextrin cavity can also enhance the PET kinetics by shortening the electron-donor distance. Thus, the PVDF-embedded film sensor based on PDI-3 and PDI-4 aggregates, and the design and fabrication methodology, showed great promise for developing effective chemosensors for trace vapor detection of aniline, with more options to be extensible for other chemical analytes through structural modification of the binding groups.

Fang's group reported on another structure design of PIs by grating PDIs onto a polymer chain through covalent linkage at one imide side (PDI-5, Figure 7a), and the PDI grafted polymer, existing in a fluorescent monomeric state in organic solvents without specific morphology, demonstrated efficient fluorescence quenching upon exposure to amine, enabling development as thin film sensor (e.g., deposited on silica gel plate) for chemical vapor detection [65]. Interestingly, the sensor showed dramatic difference in quenching

magnitude between aromatic and aliphatic amines tested under the same conditions, indicating great promise for discriminating these two classes of amines, and moreover from other common organic reagents (Figure 7a,b). While the main skeleton of polymer, poly(2-hydroxyethyl methacrylate), is optically inert as used in this sensor system (i.e., no absorption of visible light, thus no interference to the fluorescence generation and quenching), the existence of high density of 2-hydroxyethyl methacrylate side groups would impede the π-π stacking forming H-type aggregate, and consequently lead to enhancement of fluorescence that is conducive to the application as sensors. Moreover, the high density of 2-hydroxyethyl methacrylate side groups also helps enhance the adsorption of amines through hydrogen bonding interaction between amines and the hydroxyl groups on polymer backbone. Indeed, the PDI-5 demonstrated significant fluorescence quenching even upon exposure to low concentrations of aniline vapor, like 176 ppb, with detection limit projected to be 45 ppb. The film sensor also exhibited excellent sensing reversibility with rapid response and recovery as repeatedly tested between saturated vapor of aniline and clean air (Figure 7c), which both are crucial for practical applications of chemosensors, particularly in the cases of real-time, onsite detection.

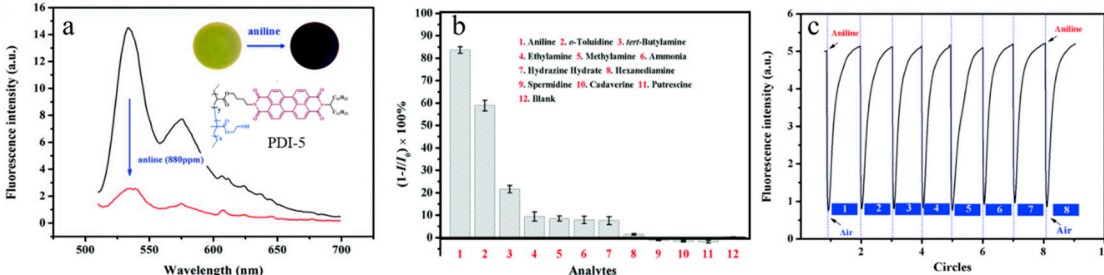

**Figure 7.** (**a**) Fluorescence spectra of a PDI-5 film deposited on silica gel plate before and after exposed to 880 ppm aniline vapor; inset shows the photographs taken from the same film under a 365 nm UV lamp before and after exposure to aniline vapor, visually indicating the effective fluorescence quenching; (**b**) Fluorescence quenching efficiency (1-I/I$_o$) measured over a PDI-5 film upon 5 s of exposure to the saturated vapor of various amines; (**c**) Fluorescence on/off response of a PDI-5 film upon 5 s of exposure to saturated vapor of aniline measured on the home-made platform ($\lambda_{ex}$ = 490 nm) [65].

Further improvement on selective detection of aromatic amines versus aliphatic amines has later been realized by Fang's group by designing and synthesizing a new PDI building block, PDI-6 (Figure 8a), which is modified with branched di(2-ethylhexyl) groups at the imide positions and two cholesterol groups at the bay positions [66]. Such structural modification resulted in improved solubility and self-assembly processability within different solvents. The self-assembly is mainly driven by the cofacial π-π stacking between the large planar and rigid conjugated structure of PDI core in concert with the Van der Waals interaction between the cholesterol groups, which combined leads to formation of 1D aggregate. Moreover, the amide structure connecting the cholesterol groups to PDI backbone may also contribute to the assembly via hydrogen bonding. Indeed, shape-defined nanofibers of PDI-6 have been fabricated from the solution self-assembly with average diameter <50 nm. When deposited onto a substrate, the nanofibers form a networked film with fluorescence emission maximum at 720 nm. Upon exposure to the vapor of amines, particularly aniline and *o*-toluidine, the nanofibers film demonstrated efficient fluorescence quenching response, paving the way to development as chemosensors. As important biomarkers of lung cancer, aniline and *o*-toluidine remain essential target for disease diagnosis through detection of exhaled breath. In addition to the high sensitivity (with detection limit ~15 ppb), the nanofibers of PDI-6 also demonstrated good selectivity and fast response (less than 1 s) towards the two aryl amines in comparison to the alkyl amines and other common organic reagents. Moreover, the sensor demonstrated good response to the simulated breath samples containing varying concentrations of aniline, for which a linear relationship between the response and concentration was obtained

(Figure 8b). The detection limit shown in this study is at least one order of magnitude lower than the concentrations (100 ppb–400 ppb) required for lung cancer detection [67].

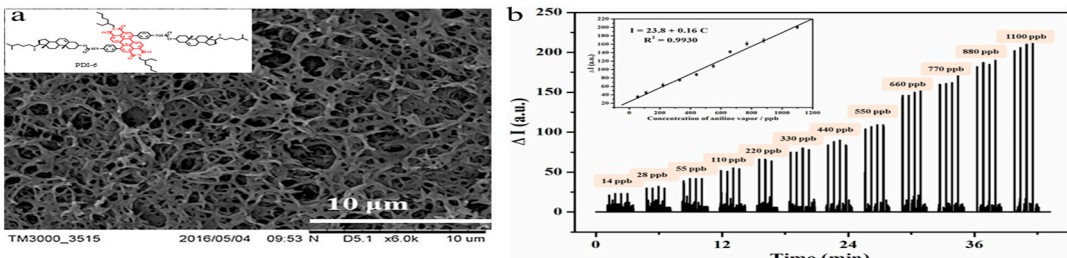

**Figure 8.** (**a**) SEM image of nanofibers (deposited as a film) assembled from PDI-6. Also shown are the molecular structure of PDI-6; (**b**) The fluorescence intensity change measured over a PDI-6 nanofiber film upon exposure to simulated breaths containing different concentrations of aniline. Inset shows the linear relationship between the fluorescence response and concentration of aniline. Note: the simulated breath samples were prepared by adding different concentrations of aniline in the breath samples collected from healthy people [66].

In another study, Fang's group designed and synthesized PDI-7 as shown in Figure 9a [68], which is modified with two cholesterol moieties at the imide positions via 2~3 units of ethyl as linker. The flexible linkage between cholesterol moiety and PDI backbone minimizes the steric hindrance, thus allowing for effective p-p stacking arrangement forming well-defined nanofibers. Indeed, shape-defined nanofibers of PDI-7 (with average diameter of 80 nm) can be fabricated simply by casting a *p*-xylene/propanol solution of PDI-7 onto a glass substrate, where the intertwined nanofibers form a three-dimensional connected network, which is conducive to gas molecule diffusion and adsorption, both essential for increasing sensor performance. The network film thus formed demonstrates strong fluorescence, which in turn can be quenched upon exposure to the vapor of amines like aniline, with a detection limit of ca. 150 ppt determined for aniline, which is close to its detection threshold as a biomarker for lung cancer diagnosis. The sensor also showed high selectivity towards aniline with relatively low interference from common organic solvents, nitroaromatic compounds and particularly phenols, implying a great potential for development as breath sensor for diagnosis of lung cancer. One drawback for PDI-7 nanofiber sensor is the slow recovery after testing with aniline, for which a full recovery requires purging with hot air for 5 min.

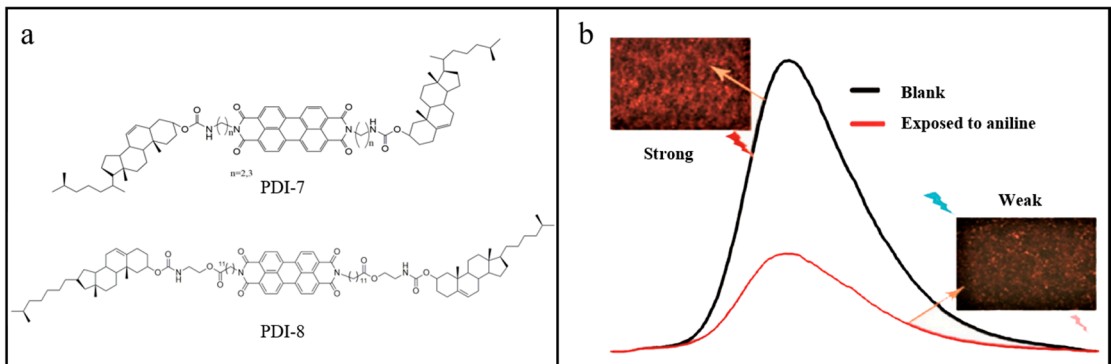

**Figure 9.** (**a**) Molecular structures of building blocks PDI-7 and PDI-8; (**b**) Fluorescence spectra of nanofibers film of PDI-7 in the absence and presence of aniline vapor, with inset showing the photographs taken on the corresponding films [68,69].

Similar to PDI-7, another building block PDI-8 (Figure 9a) was also synthesized and characterized by the Fang group [68,69]. Especially, the presence of ester and amide moieties within the linker plays essential role (e.g., hydrogen bonding) in controlling the molecular assembly. PDI-8 possesses an even longer linker (11 units of ethyl) between the cholesterol moiety and PDI backbone. Larger fibers (with average diameter around 100

nm) were fabricated from PDI-8 via chloroform/ethanol phase-transfer self-assembly, for which the morphology of aggregation was mainly controlled by the columnar stacking of the PDI units in concert with the hydrogen bonding and Van der Waals interactions between the side chains. Although showing relatively weak response to aniline vapor, the film of PDI-8 nanofibers (also in the morphology of fibril network) exhibited super-sensitive fluorescence quenching response towards gaseous N-methyl-phenethylamine (MPEA), which is a common simulant of N-methamphetamine (MAPA), one of the most popular illicit drugs used throughout the world. Furthermore, the sensing response was found to be highly selective towards MPEA with minimal interference from other amines, common organic solvents, water, apple pomace, etc. Better than PDI-7, the PDI-8 film sensor can be recovered after testing simply by purging ambient air for 5 min.

In addition to the common nanostructures like nanorod or nanowire mentioned above, some other unique structures like nanocoils have also been fabricated recently by Che and co-workers based on specially designed building blocks of PIs, such as PDI-9 (Figure 10a) [47]. Formation of the coil structure is due to the J-type helical π-stacking geometry (Figure 10a), which favors the growth of assembly along the coil direction. TEM imaging (Figure 10b) clearly revealed the nanocoil architecture [47]. The nanocoils of PDI-9 are strongly fluorescent with quantum yield of 25%. This, in combination with the inherent interior porosity and high surface area, makes the nanocoil materials ideal candidate for development as fluorescence sensor for vapor detection of amines. Indeed, as tested in this study, the film of PDI-9 nanocoils demonstrated highly sensitive fluorescence quenching when exposed to the vapor of aniline and phenethylamine, with detection limits estimated as 0.8 and 3 ppt, respectively. The sensing can be simply recovered through heating the film at 60 °C for 10 min. Remarkably, the nanocoil sensor showed much higher sensitivity to aromatic amines over alkyl amines (detection limits at ppb level). Moreover, little sensor response was observed for the vapor of common organic solvents. This indicates strong detection selectivity towards aromatic amines, and such selectivity is mainly due to the superior binding strength of aromatic amines that are capable of both π-π interaction and electron donor-acceptor interaction with the PDI backbone (in comparison to the case of alkyl amines that bind to PDI only through the latter).

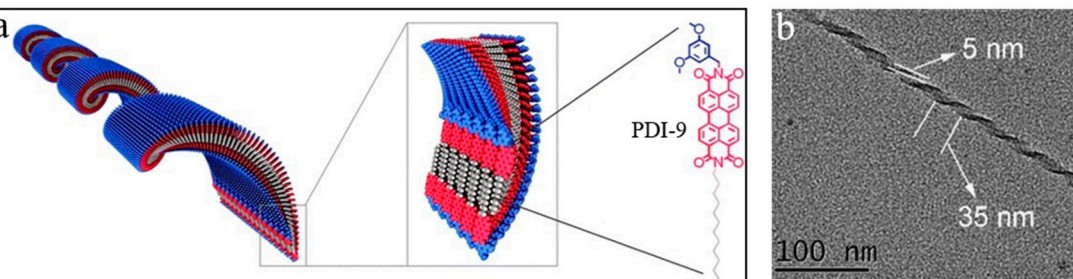

**Figure 10.** (**a**) Schematic illustration showing the molecular self-assembly of PDI-9 leading to formation of nanocoils; (**b**) TEM image of a single nanocoil formed from PDI-9 [47].

### *4.2. Benzene Homologs*

Compared with organic amines, benzene homologs represent a series of major VOCs in polluted air, which can pose a serious threat to human health and the environment. Particularly, the BTEX (standing for benzene, toluene, ethylbenzene, *o*-xylene, *m*-xylene and *p*-xylene as listed in Table 1) are more volatile, more toxic and have much wider distribution in the environment. Some BTEX are also found in the exhaled breath of cancer patients, and thus can be considered as a biomarker for cancer diagnosis [70,71]. Therefore, the research on the gas-phase detection of benzene homologs has attracted increasing interest in the field of PIs-based optical chemosensors, though the detection of these VOCs remains challenging mainly because of their rigid (less flexible) structure, chemical inertness (regarding redox interaction or reaction), and the high similarity among

the homologs that makes it difficult to distinguish them. To address these challenges in order to reach sufficient detection specificity, special effort has to be put on the elaborate structural design of PIs, as well as the fabrication and optimization of the nanomaterials.

　　Fang et al. reported on an interesting asymmetric PDI building block molecule, PDI-10 (Table 2), which is modified with a long alkyl chain at one imide position, and a rigid, bulky and shape-persistent pentiptycene moiety at the other end of PDI backbone [24]. Such substitution modifications provide significant increase in the solubility of PDI-10 in common organic solvents, which is conducive to solution processing for material fabrication. The increased solubility is mainly due to the nonplanar steric hindrance caused by the bulky pentiptycene unit, which in turn inhibits the intermolecular π-π stacking, thus resulting in an increase in fluorescence of the assembly of PDI-10 (with fluorescence quantum yield of 12.2% determined for the film of PDI-10). Moreover, the nonplanar steric configuration of the pentiptycene unit also helps generate molecular channels in the aggregate state of PDI-10 when deposited as a film on a substrate (e.g., silica-gel plate) via drop-casting of a chloroform solution. The molecular porosity thus generated is conducive to enhancing the diffusion and absorption of hydrophobic molecules like BTEX via possible capillary condensation effect and solvophilic effect. As a consequence, PDI-10 film-based fluorescent sensor, as tested with a home-made vapor detection platform (shown Figure 2), demonstrated unprecedented sensing performance towards BTEX vapors as evidenced by the fast response (in seconds), high sensitivity and reliable discrimination of BTEX from other aromatic hydrocarbons including those (e.g., mesitylene) with similar chemical structure (Table 1), and the potential interferences from alkyl amines, common organic solvents, water, etc. Specifically, the detection limits for the 6 BTEX (benzene, toluene, ethylbenzene, *o*-xylene, *m*-xylene and *p*-xylene) were determined as 9.2, 2.7, 1.9, 0.2, 0.4 and 0.4 ppm, respectively. Except for benzene, the detection limits for the BTEX VOCs are significantly lower than the NIOSH recommended long-term exposure limits. The observed sensing is fully reversible with the recovery time (usually less than 3.5 min) depending on the boiling point and saturated vapor pressure of analytes (Table 1), as well as air-blowing rate. Moreover, the film of PDI-10 was proven chemically stable under photo-excitation, even after 300 cycles of repetitive illumination tests, which indicates great potential for deployment in real applications.

**Table 1.** Chemical structures and important physical parameters of BTEX and mesitylene [24].

| Analytes | Benzene | Toluene | Ethylbenzene | *o*-Xylene | *m*-Xylene | *p*-Xylene | Mesitylene |
|---|---|---|---|---|---|---|---|
| Chemical structure |  |  |  |  |  |  |  |
| Boiling point (°C) | 80.1 | 110.8 | 136.2 | 144.4 | 139.1 | 138.4 | 164.8 |
| Vapor Pressure ([mm Hg], 25 °C) | 95.2 | 28.4 | 9.6 | 6.7 | 8.4 | 8.8 | 2.5 |

### 4.3. Organic Peroxides

　　Different from the gaseous analytes like amines and benzene homologs discussed above, organic peroxides are much more chemically active and thus more toxic to organism health and hazardous to environmental safety. Among them, triacetone triperoxide (TATP) and diacetone diperoxide (DADP) represent one type of the most common organic peroxides mainly due to their illicit use as high explosives in recent years, and the precursor used in the synthesis of these two peroxides are acetone and hydrogen peroxide, which are both common chemical reagents and commercially available [72]. Thus, development of efficient chemosensor technologies that are sensitive, selective, quick and non-invasive remains imperative for real-time onsite detection or screening of acetone and related peroxide ex-

plosive, and especially of great interest to public safety. Porous fluorescent films, especially those fabricated from entangled nanofibers, provide both large surface area for absorption and open porosity for expedient molecule diffusion. The combination of these two features makes nanofibril films ideal sensor materials for vapor detection of explosives [73].

Recently, Fang's group developed a unique fluorophore based on PMI structure, PMI-2 (Figure 11a), which can be fabricated as a fluorescent film by drop-casting a THF solution onto a silica-gel plate, followed by air drying [74]. The non-planar geometry of PMI-2 inhibits the H-type cofacial stacking, thus resulting in an aggregate of PMI-2 with significant fluorescence. Moreover, the interstitial formed by the non-planar configuration of two PMI units enables strong adsorption of gaseous analytes, mainly due to capillary condensation and the solvation effect (Figure 11b). Particularly, with the introduction of a four-coordinate monoboron unit (which possesses $sp^3$ form of molecular orbital), the PMI-2 molecule exhibits a tetrahedral configuration, which could further inhibit the dense packing of PMI backbones, thereby producing porous internal structures at both the molecular level and aggregate state. The internal porosity thus formed, in conjunction with the intrinsic porosity of silica gel substrate, is essentially conducive to the diffusion and adsorption of gas analytes, and thus enhancement in sensor performance, just like the case of PDI-10 as discussed above. Indeed, the PMI-2 film showed excellent fluorescence sensing towards acetone vapor with response time less than 2 s and recovery time within 10 s, and an experimental detection limit lower than 50 ppm. The sensor also showed nearly perfect reversibility and minimal interferences from vapors of common organic solvents, air, water, $H_2O_2$, odorous toiletries, fruits and dirty clothes or other types of explosives like 2,4,6-trinitrotoluene (TNT) and 2,4-dinitrotoluene (DNT), etc. The same sensor can also be used for vapor detection of TATP and DADP with experimental detection limits lower than 30 and 50 mg, respectively. Moreover, the PMI-2 film demonstrated great stability as tested, with no observable degradation in the sensor performance after more than 30 days of continuous use. Such strong robustness is especially desirable for remote or onsite detection, where continuous monitoring is often required.

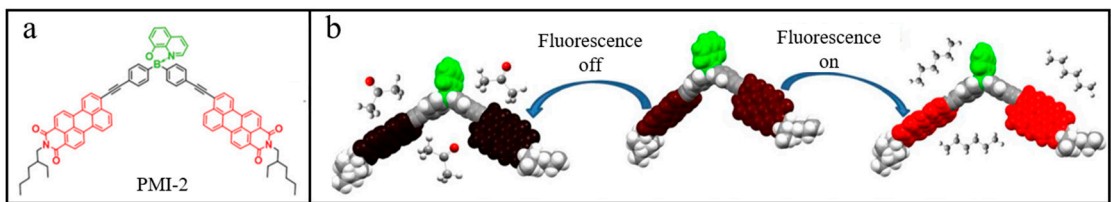

**Figure 11.** (**a**) Molecular structure of PMI-2; (**b**) Schematic diagram for the process of capillary condensation and solvation effect enabling absorption of vapor analytes within the film of PMI-2 [74].

### 4.4. Phenols and Nitroaromatics

Phenols and nitroaromatics represent electron-rich and electron-deficit molecules, respectively, which both are redox active for electron transfer interaction with chemosensors leading to efficient fluorescence quenching or enhancement. Unlike the highly volatile analytes mentioned above, the vapor pressure of these two types of VOCs are not really high under ambient conditions, which is consistent with their boiling points, e.g., 181.9 °C for phenol, 210.9 °C for nitrobenzene (NB). The pollutants of these compounds are mostly present in water environment, rather than in air. That's why many chemosensors have been developed for water phase detection of phenols and nitroaromatics. Phenols and nitroaromatics are high toxic, and often persistent to natural degradation in environment [75]. Nonetheless, vapor-phase detection of these VOCs still remains of strong interest because it provides direct monitoring and assessment of the air quality, and more importantly, vapor sensing enables quick, non-invasive detection and this is especially crucial for detecting nitroaromatics like TNT and DNT that are often used as explosives. In this section we

discuss some recent research advancement in fluorescence chemosensors for vapor phase detection of phenols and nitroaromatics.

As depicted in Figure 12, Fang Yu and his co-workers reported on an interesting supramolecular ensemble combining one PDI molecule, PDI-11 (modified with ammonia salt at the two imide positions), and two molecules of cholesterol-functionalized calix[4]pyrrole (CCP) [76]. Reversible assembly and disassembly between the two compounds would be easily achieved by deprotonation or protonation of the carboxylic acid moiety. The intermolecular assembly is due to hydrogen bonding between the carboxylic group of PDI-11 and the amines on CCP, and such supramolecular assembly can be dissociated upon presence of phenols or nitroaromatics, leading to dramatic fluorescence quenching, a modulation that can be used for chemosensors. When dissolved in an ethanol solution, the fluorescence of CCP-PDI-11 assembly was gradually quenched upon addition of increasing concentrations of phenol. The observed quenching was attributed to the direct electron donor-acceptor interaction with the PDI backbone as illustrated in Figure 12. Effective fluorescence quenching was also observed for TNT in ethanol solution, which was otherwise attributed to the strong association of TNT within CCP (controlled by both strong hydrogen bonding and electron donor-acceptor interaction), which in turn results in formation of stacking aggregation of PDI backbones and thus quenching of fluorescence (with a detection limit $\sim$80 nM). More interestingly, the fluorescence sensing for phenol was also observed in the film state of the CCP-PDI-11 assembly, with a detection limit lower than 1 ppb. This study shows the possibility of an alternative way to design chemosensors based on composite structure incorporating multiple functional molecules, which may help improve the sensor performance regarding both sensitivity and selectivity.

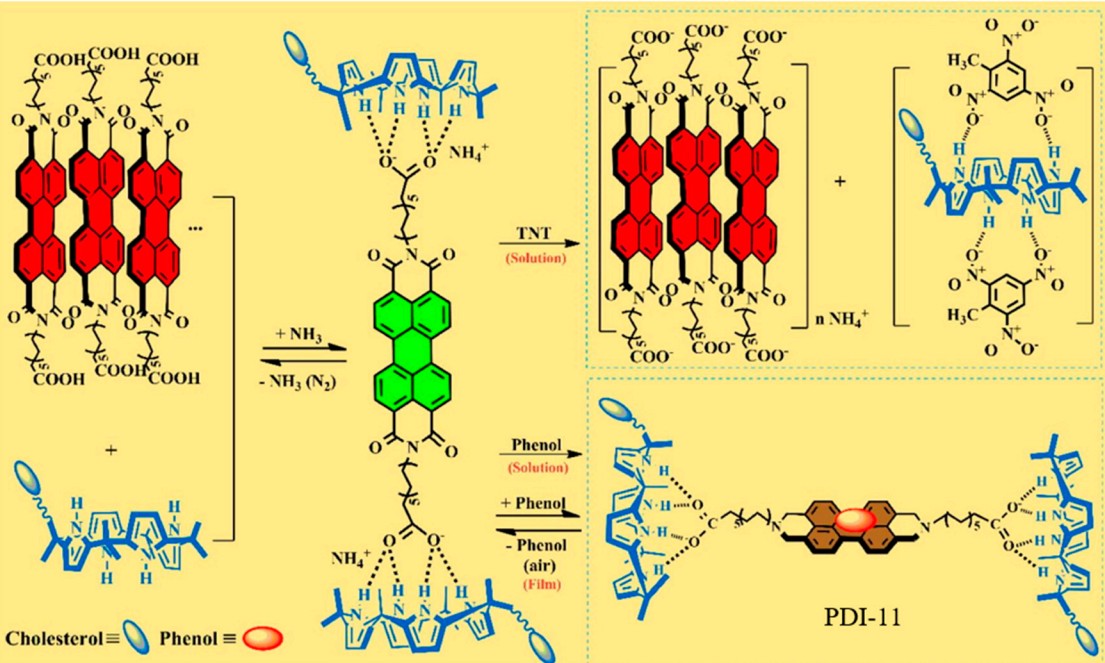

**Figure 12.** Schematic representation of the assembly and disassembly of a fluorescent supramolecular ensemble of calix[4]pyrrole with PDI-11, and its interaction with TNT and phenol in two different modes leading to turn-on of PDI aggregate fluorescence and quenching of fluorescence of PDI molecules. Note: the former is irreversible, but the latter could be recovered by purging of $N_2$ or air [76].

The PDI-calix[4]pyrrole sensor illustrated in Figure 12 implies the potential of developing composite sensory materials that may enable detection of both electron-deficit nitroaromatics and electron-rich amines, though the reported sensor demonstrated similar fluorescence quenching response for both the two types of analytes [76], limiting the capability of detection selectivity. To further improve the multi-target sensing with nec-

essary selectivity, Ajayaghosh and co-workers [77] designed and developed a dual-mode fluorescence "turn-on" sensor that can selectively detect nitroaromatics and aromatic amines based on different colors of the turned-on fluorescence emission in (Figure 13a,b). The sensor material is based on supercoiled supramolecular polymeric fibers fabricated from solution phase co-assembly of $C_3$-symmetric oligo(p-phenylenevinylene) ($C_3$OPV) and $C_3$-symmetric PDI molecule (PDI-12, Figure 13a), which function as electron donor and acceptor, respectively. When mixed at a 1:1 molar ratio, $C_3$OPV and PDI-12 form self-sorted, shape-defined nanocoil structure taking advantage of self-sorting stacking at the molecular level and electronic donor-acceptor interactions at the macroscopic level. Due to the efficient PET between $C_3$OPV and PDI-12 coils, the supercoiled composites as fabricated are nonfluorescent, providing an ideal zero background for turn-on sensing. Upon exposure to the vapor of aromatic amines like *o*-toluidine, a green emission of $C_3$OPV part emerged due to the competitive binding between *o*-toluidine and PDI, which in turn frees the OPV part from PET quenching with the same PDI (in Figure 13c). A similar but opposite response occurs as exposure to the vapor of nitroaromatic molecule like 2-nitrotoluene (strong electron acceptor) frees the PDI part, thus turning on the red fluorescent emission intrinsic of PDI. The overall sensing sensitivity was found to be dependent on the electron accepting or donating capability of analytes, as well as their vapor pressures under ambient conditions. The reported dual-mode chemosensor represents a unique, innovative approach to realizing sufficient detection selectivity that is usually demanded in practical applications wherein diverse types of unknown analytes may exist.

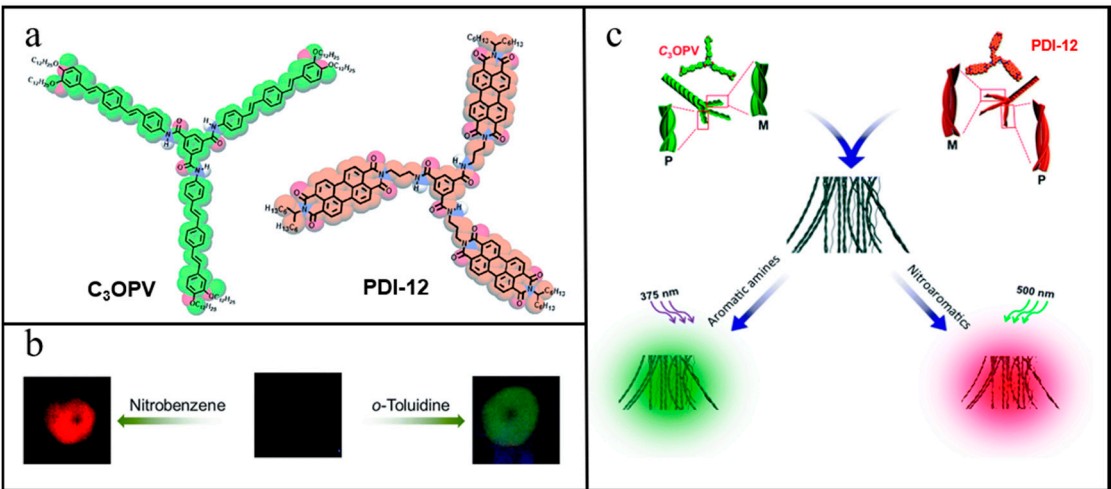

**Figure 13.** (**a**) Chemical structures and molecular models (shown in color) of $C_3$OPV and PDI-12; (**b**) Photographs showing the fluorescence of a film fabricated from 1:1 mixture of before and after exposure to nitrobenzene and *o*-toluidine vapor, leading to turn-on of the fluorescence of PDI-12 and $C_3$OPV, respectively; (**c**) Schematic illustration of the fluorescence 'turn-on' mechanism of the self-sorted fibers of a 1:1 mixture of $C_3$OPV and PDI-12 upon exposure to two different types of VOCs, electron donor like amines and electron acceptor like nitroaromatics [77].

In another dual-mode approach (but based on fluorescence quenching), Asha SK and co-workers [78] developed a polymer nanocomposite mixed with two fluorophores that respond to different types of analytes, electron donor vs. acceptor. Specifically, polystyrene nanobeads have been synthesized that incorporate both PDI and oligo (p-phenylenevinylene) (OPV) fluorophores through selective functionalization with carboxy and amine groups as illustrated in Figure 14. Figure 14a shows the molecular structure of the PDI (PDI-13) and OPV oligomer used in the study. The nanobeads as synthesized are in a size range of 70–180 nm, and showed near white emission under UV light that combines the fluorescence from both PDI and OPV. Upon exposure to the vapor of specific electron deficient nitroaromatics like nitrobenzene, the emission of OPV gets selectively quenched (via PET from OPV to nitroaromatic), resulting in yellow emission from the

nanobeads under UV radiation (which is mostly dominated from the fluorescence of PDI) (Figure 14b–f). In contrast, if exposed to the vapor of electron rich analytes like amines (e.g., *o*-toluidine), the emission of PDI will be selectively quenched (via PET from amine to PDI as commonly observed by many others as mentioned above), resulting in blue emission from the beads mostly contributed by the fluorescence from OPV part (Figure 14b–d). Such a dual-color change from white baseline provides enormous options to distinguish different analytes with varying electron donating/accepting capability via CIE colorimetry analysis as shown in Figure 14e. Moreover, the surface functionalization with −COOH and −NH$_2$ groups facilitates the adsorption of amines and nitroaromatics through hydrogen bonding (Figure 14f). This, in combination with the large open surface area of nanobeads, enables fluorescence sensing with both high sensitivity and fast response. The nanobeads also demonstrated good recyclability after 8 cycles of testing.

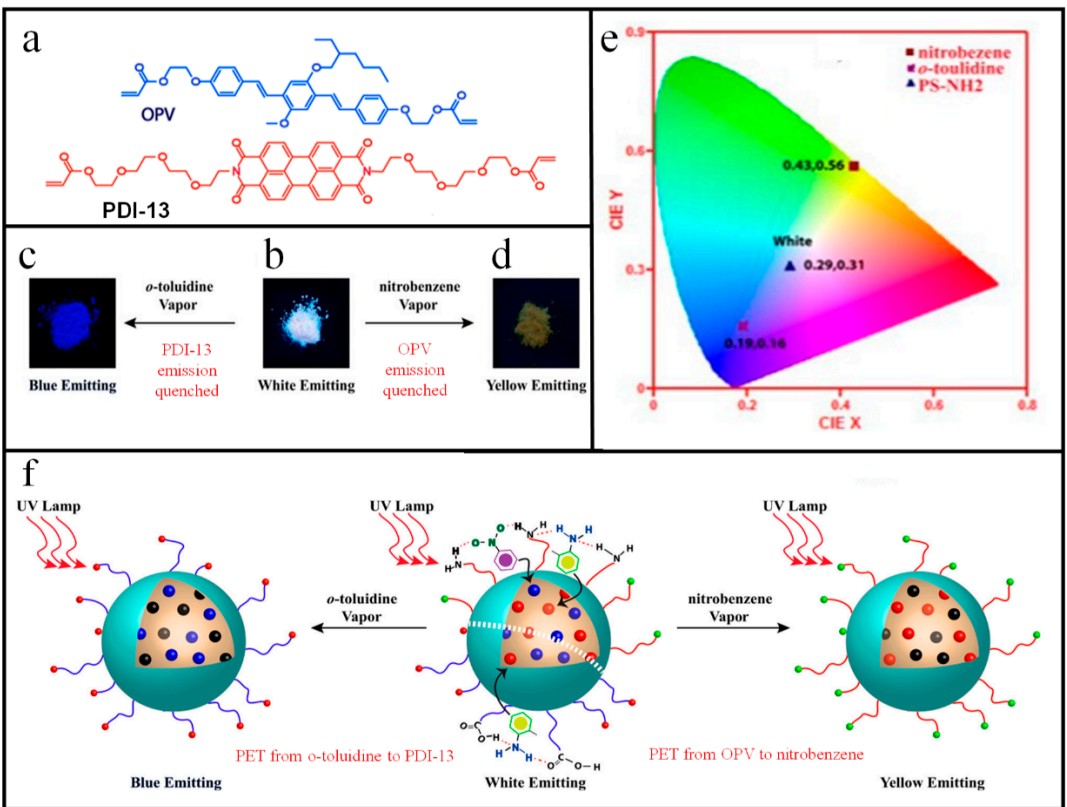

**Figure 14.** (**a**) Molecular structures of OPV and PDI-13; Photographs taken on a sample of polystyrene nanobeads containing both OPV and PDI-13 under UV lamp before (**b**) and after exposure to *o*-TD vapor (**c**) and NB vapor (**d**,**e**) CIE coordinate diagram for the polystyrene nanobeads showing the color tuning and sensing of both *o*-TD and NB vapor; (**f**) Possible quenching mechanism by the nitro-organics and amine-organics vapors through respective PET process [78].

### 4.5. Optical Chemosensors Array

As mentioned above, most of the chemosensors reported and developed thus far are based on a single sensor component or material that can target a certain class of analytes (e.g., amines vs. nitroaromatics), but hardly distinguish between individual gases or VOCs (e.g., methylamine vs. ethylamine, or alkyl amines vs aromatic amines). Although significant efforts have been made to improve the molecular design, aiming to enable specific detection with just one sensor material or device, there have been very few chemosensors that can selectively detect and identity a specific compound—the chemical similarity among the analogous compounds always causes some extent of response signals. Nonetheless, it still remains impractical to use such single-component sensor for detection in the cases where there may exist unknown analytes, or where the target analytes are

present with other interferences from the local environment. To address this challenge, an alternative approach to better detection specificity is through an array of sensors, for which the cross-reactive responses from all the sensors (e.g., different color changes, intensity changes, response/recovery time, etc.) are collected and incorporated in algorithm analytics, thus enabling differential sensing and distinction between different analytes based on pattern recognition [79]. Building upon the primary detection selectivity already optimized for the individual sensor component as discussed in above sections, an array of the sensors would provide significant improvement in detection selectivity, just like the excellent performance of the mammalian olfactory system, wherein the large number of olfactory receptors (sensors) work as a cooperative array that can generate specific patterns for different odors or mixtures [80].

In recent years, optical chemosensors have drawn increasing interest in design and construction of sensor arrays, due to their diverse of response signals and flexibility in both materials and device processing. Arrays of chemosensors have been proven capable of detecting not only chemical analytes (VOCs, explosives, ions, biological macromolecules, etc.), but also the physical stimulants (force, stiffness, etc.). For instance, Yu et al. [81] recently developed a new type of PDI-based fluorescence sensor array, which can detect and distinguish 11 metal ions, including $Cu^{2+}$, $Fe^{3+}$, $Pb^{2+}$ and $Zn^{2+}$, through differential sensing. It is usually difficult to distinguish the metal ions with just one type of sensor. This is particularly true for the ions with similar chemical properties, mainly due to cross sensing interference. In general, a sensor array incorporating multiple probes for multiple analytes provides much richer sensing response information than the traditional single component probe [82], thus enabling high capacity of differential sensing based on appropriate algorithm analysis such as the advanced machine learning methodologies. With such an array approach, the overall selectivity can be enhanced (when needed) through increasing the number of sensor components (chromophores) and/or recognition groups of a series of probes, thereby removing the need for complicated, sophisticated synthesis of a sensor molecule to bind or interact with a specific target analyte, which is often a very challenging task.

As another successful example of chemosensor array, three pyrene-containing fluorophores (Py-PE, Py-CB-Ph, and PDI-14, shown in Figure 15) have recently been synthesized and developed as efficient fluorescence film sensors, and incorporated into an array by Fang's group [83]. The array demonstrated highly efficient detection and discrimination of 15 chemicals selected from the broad range of targets of public safety, health and environment interest, including five typical explosives (2,4,6,8,10,12-hexanitro-2,4,6,8,10,12-hexaazaisowurtzitane, triacetone triperoxide, ammonium nitrate, 1,3,5-trinitroperhydro-1,3,5-triazine, black powder), five illicit drugs (methamphetamine, magu, ecstasy, caffeine, phenobarbital), and five VOCs (acetone, toluene, chloroform, methanol, diethyl ether). The differential sensing power of the array lies mostly in the diverse fluorescence spectral modulations of the three sensor molecules as evidenced in both the solution phase and aggregate state in the film (Figure 15b). Specifically, Py-PE remains robust fluorescent in both solution and solid films, though the fluorescence in the latter gets decreased and red-shifted to some extent. In contrast, Py-CB-Ph possesses a typical push-pull structure, which results in significant aggregation-induced yellow emission, greatly enhanced upon drying out from a cast solution forming solid state film. Every sensor as fabricated as a film exhibited characteristic fluorescence responses to the same analyte; incorporating all the responses from the three sensors would produce a pattern that contains sufficient information for discriminating different chemicals. More interestingly, PDI-14 was designed to get a large Stokes' shift allowing for selective excitation (355 nm) of the Py unit, which generates strong green emission from the PDI part, rather than the pyrene part, likely due to a complete Förster energy transfer process. This provides a unique way to expand the sensor spectral response by employing various fluorophores, thus making it possible to further enhance the differential sensing capacity of an array. A conceptual detector platform was constructed based on a sensor array incorporating the three chemosensors, which was

proven successful for pilot in-field testing, particularly against the potential interference from the vapor of water, toiletries, fruit, dirty clothes and other environmental species.

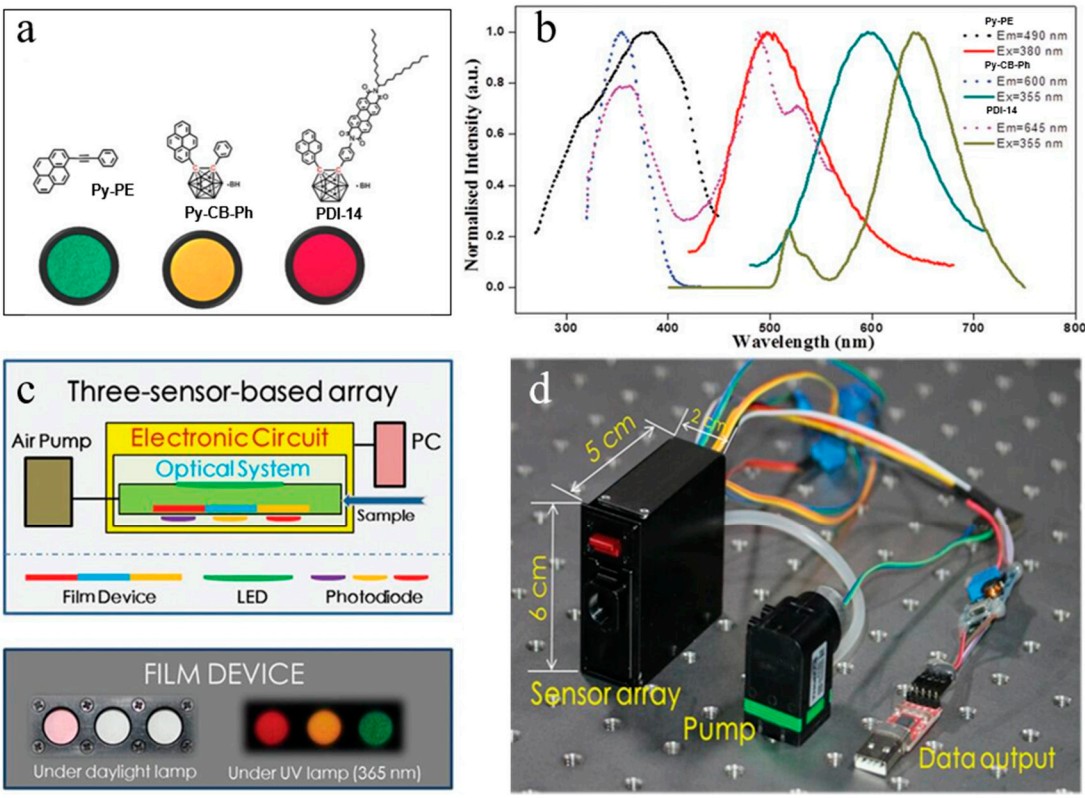

**Figure 15.** (**a**) Molecular structures of the three synthesized fluorophores (Py-PE, Py-CB-Ph, and PDI-14) and photographs taken over the corresponding films under UV light (365 nm) showing different emission colors; (**b**) The excitation and emission spectra of the films fabricated from the three molecules; (**c**) Schematic illustration of the detector platform based on an array of sensors (films) fabricated from the three molecules, with the daylight and fluorescence photographs of the three films shown I the bottom panel; (**d**) Photograph of the real detector system consists of different parts [83].

As evidenced by many recent studies, the fluorometric and colorimetric sensors, when integrated as arrays, can become more powerful and successful in detecting and identifying different or mixed analytes. PIs may find increasing application in these applications regarding its high adaptability and flexibility in structure design and optical/electronic property modulation. Especially in aggregate state, the color of PIs (as well as the emission) can be widely modulated, covering the entire visible region by tuning the intermolecular arrangement in concert with the molecular structure modification (at both the imide and bay positions as discussed above). We expected that more research results and discoveries of PIs will come out in next a few years on the design and synthesis of new molecules that can be directly developed as efficient fluorometric or colorimetric sensors (even simply with naked-eye readout). The diverse availability of PIs provides more options for the assembly of arrays that can be optimized by selecting different sets of sensor materials to target some specific analytes or environment scenarios to be monitored. Moreover, PIs have been proven ideal building blocks for self-assembly, forming various 1D nanostructures like nanofibers, which are highly suited for fabrication into chemosensors in small size and miniaturized system as successfully practiced and commercialized by Vaporsens. With the recent advancement in nanofabrication and device integration with soft materials and films, it is no surprise to see more employment of PIs-based chemosensors in construction of arrays and detectors, particularly those in flexible or wearable formats to be more favorable for real-time, on-site and portable analysis [54,84,85].

## 5. Conclusion and Prospectives for Future Work

In conclusion, as shown in Table 2, PIs-based optical chemosensors have been proven to be successful in the detection of toxic chemicals, VOCs, explosives and other analytes in the gas-phase environment with high selectivity and sensitivity, as well as sufficient photochemical stability and reversibility. Summarized in this review is not only the various design rules and strategies of sensor molecules and nanostructures fabricated there from, but also the different sensing modes (as well as multimodal and array systems) and the diverse analytes that can be detected. To the regard of difficulty level of fabrication and limited availability of sensor materials, colorimetric sensors have significantly less studied than the fluorometric counterparts, particularly for the sensors based on PIs. In addition, the diverse options of PIs regarding molecular structure and self-assembly processability make developing PIs as fluorometric sensors working alone or in an array a great opportunity. Incorporation of multiple sensors in an array helps identify multiple analytes simultaneously and discriminate a specific analyte from a mixture of analogues. With the currently available nanoscale fabrication technologies and the advanced algorithm analysis, the sensor array approach may provide much enhanced detection capacity regarding both selectivity and discrimination from a mixture of multianalytes, which mimic and will likely surpass the sniffing capacity of mammalian olfactory system.

**Table 2.** Comparison of PIs based optical chemosensors for different vapor analytes.

| PIs | Molecule Structures | Analytes | Sensing Signals | Response Times | Detection Limits (ppb) | Reversibility | Ref. |
|---|---|---|---|---|---|---|---|
| PDI-10 |  | *o*-Xylene | FL on-off | 3.4 s | 200 | yes | [24] |
| PDI-1 |  | ethylene diamine | FL on-off | <5 s | < 590 | yes | [25] |
| PMI-1 |  | aniline | FL on-off | 0.32 s | <0.2 | yes | [41] |
| PDI-9 |  | aniline | FL on-off | – | 0.0008 | yes | [47] |
| PDI-4 |  | cadaverine; putrescine | FL on-off | <1 h | 1.2; 2.6 | yes | [61] |
| PDI-2 |  | aniline | FL on-off | 10 s | 33,000 | – | [63] |
| PDI-3 |  | aniline | FL on-off | 10 s | 80 | yes | [64] |
| PDI-5 |  | aniline | FL on-off | <1 s | 45 | yes | [65] |

**Table 2.** *Cont.*

| | | | | | | | |
|---|---|---|---|---|---|---|---|
| PDI-6 | | aniline | FL on-off | <1 s | 15 | yes | [66] |
| PDI-7 | | aniline | FL on-off | <3 min | 0.15 | yes | [68] |
| PDI-8 | | MPEA | FL on-off | 50 s | 5.5 | yes | [69] |
| PMI-2 | | acetone | FL on-off | 2 s | 50,000 | yes | [74] |
| PDI-11 | | TNT; phenol | FL off-on &color | – | 1792; <1 | yes | [76] |
| PDI-12 | | NB | FL off-on | – | – | – | [77] |
| PDI-13 | | *o*-TD | FL off-on | 30 min | 6.9 | yes | [78] |
| PDI-14 | | explosives; illicit drugs; VOCs | FL on-off | – | – | – | [83] |

**Author Contributions:** S.C. and L.Z. formulated the review scope and topic, and helped revise the manuscript. M.Z. organized the literature and drafted out the manuscript. J.S., C.L. and Q.T. helped in preparation of figures, and were also involved in discussions that helped improve the writing. C.W. provided comments and suggestions helping improve the manuscript. All authors have read and agreed to the published version of the manuscript.

**Funding:** The financial support from the Scientific Research Fund of Shaanxi University of Science and Technology were gratefully acknowledged. S.C. thanks the support from China Scholarship Council (No. 201808360327).

**Conflicts of Interest:** The authors declare no conflict of interest.

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
