# Peer review of "Perylene Imide-Based Optical Chemosensors for Vapor Detection"

_chemosensors, doi:10.3390/chemosensors9010001_

Round 1
Reviewer 1 Report
This review by Zhang and co-workers summarises the use of perylene diimide as a optical chemosensor for vapor-phase detection of a wide variety of organic compounds which has human and environmental impacts. I believe this article is well written with some very minor errors as listed in the attached file.

Reviewer 2 Report
Zhang et. al reported review is well written, it can be acceptable with minor corrections mentioned below.
1) Authors need to keep substitutions oriented morphological changes of aggregates (like chiral cholesterol and aromatic groups can induce different types of nano-structures etc. !!!).
As the entire paper describes aggregates of various PIs derivatives along with its sensing performance of amines and phenols, it is highly desirable to incorporate the morphological changes based on substitutions at specified positions.
2) In the last section Table 1 comparison table, solid/solution state quantum yield should be reported.
Reviewer 3 Report
Zhang et al outlined a systematic review on the development of chemosensors for vapor detection using perylene imide-based chemistry. This work was based on the previous works published by the same group. I found the quality of this review is satisfactory and publishable. However, I also question its novelty compared to the other review published by the same group of authors (Sensors 2020, 20, 917; doi:10.3390/s20030917, Perylene Diimide-Based Fluorescent and Colorimetric Sensors for Environmental Detection). In particular, certain paragraph and sentences are extremely similar or even the same. Although one of the work focused on using perylene diimide to detection environmental pollutants, it is debatable whether the field needs these two similar reviews on closely related topics. The authors are suggested to emphasize more on this point prior to publication in this journal
Round 2
Reviewer 3 Report
The revised manuscript has addressed my previous concerns.